# Targeting the C-Terminal Domain Small Phosphatase 1

**DOI:** 10.3390/life10050057

**Published:** 2020-05-08

**Authors:** Harikrishna Reddy Rallabandi, Palanivel Ganesan, Young Jun Kim

**Affiliations:** Department of Medicinal Biosciences and Nanotechnology Research Center, Konkuk University, Chungju 27478, Korea; harikrishna@kku.ac.kr (H.R.R.); palanivel67@gmail.com (P.G.)

**Keywords:** CTDSP1, drug design, allosteric docking, ensemble docking

## Abstract

The human C-terminal domain small phosphatase 1 (CTDSP1/SCP1) is a protein phosphatase with a conserved catalytic site of DXDXT/V. CTDSP1’s major activity has been identified as dephosphorylation of the 5th Ser residue of the tandem heptad repeat of the RNA polymerase II C-terminal domain (RNAP II CTD). It is also implicated in various pivotal biological activities, such as acting as a driving factor in repressor element 1 (RE-1)-silencing transcription factor (REST) complex, which silences the neuronal genes in non-neuronal cells, G1/S phase transition, and osteoblast differentiation. Recent findings have denoted that negative regulation of CTDSP1 results in suppression of cancer invasion in neuroglioma cells. Several researchers have focused on the development of regulating materials of CTDSP1, due to the significant roles it has in various biological activities. In this review, we focused on this emerging target and explored the biological significance, challenges, and opportunities in targeting CTDSP1 from a drug designing perspective.

## 1. Introduction

RNA polymerase II (RNAP II) is the crucial component of a transcription apparatus, and orchestrates the post-transcriptional regulation and modification of mRNA. The largest subunit of RNAP II is the C-terminal domain (CTD), which is essential for the recruitment and assembly of transcription complexes. CTD contains consensus tandem heptapeptide (Y_1_S_2_P_3_T_4_S_5_P_6_S_7_) repeats, which vary from 26 to 52 from yeast to humans, respectively [1,2]. CTD phosphatase 1 (CTDP1/FCP1) is the first member of a C-terminal domain phosphatase (CTDP) family, which was identified by dephosphorylating the Ser2 of heptapeptide [3,4]. Later studies revealed that CTD small phosphatase 1 (CTDSP1/SCP1) preferentially reverses the phosphorylation of the Ser5 in consensus peptide repeats of RNAP II CTD [5,6]. CTD small phosphatase 2 (CTDSP2/SCP2) and CTD small phosphatase-like (CTDSPL/SCP3) were identified as structurally and functionally closer paralogs of CTDSP1 [5,7]. CTDSP1 hydrolyzes the phosphoryl group on the Ser5 of CTD through the conserved catalytic site of DXDXT/V and Mg^2+^ ions, which is highly conserved throughout CTD small phosphatases (CTDSP).

The catalytic activity of CTDSP1 extends its role into various vital biological activities, and has made it an anticipated drug target [8]. CTDSP1’s novel signature mechanism is essential for numerous signaling pathways and cellular activities, such as neuronal gene silencing, cell cycle regulation, and regulation of certain cell signal transductions. Repressor element 1 (RE-1)-silencing transcription factor/neuron-restrictive silencer factor (REST/NRSF) complex is responsible for silencing the neuronal genes, and some studies have illustrated that CTDSP1 is co-precipitated with the REST complex in non-neuronal cells [9,10]. A recent study suggested that CTDSP1 controls the stability of the REST complex through dephosphorylation [11]. Phosphorylated receptor-regulated small mothers against decapentaplegic (SMAD) proteins (R-SMADs) play a significant role in the orchestration of transforming growth factor β (TGFβ) signal transduction and bone morphogenetic protein (BMP) signaling. The R-SMADs could be dephosphorylated by the nuclear CTDSP1 and attenuate the BMP signal at the required level [12]. Further studies have demonstrated CTDSP1’s role in cell cycle arrest at the G1/S transition phase by C-Myc mediated dephosphorylation of the retinoblastoma protein (RB) [13,14]. A new study has suggested that cell division associated 3 protein (CDCA3) could also be an enzymatic substrate for CTDSP1 [15]. Apart from these known functionalities, a recent study related to cancer invasion exhibited the exciting results that CTDSP1 dephosphorylates Twist-related protein (TWIST) and regulates cancer cell migration [16]. Given all these characteristics, we are seeking to prove that CTDSP1 can be considered as a potential drug target.

There are a few challenges in targeting CTDSP1 despite its potential. In previous studies [17,18,19], researchers tried to identify chemical inhibitors targeting human CTDSP1 and emphasized structure-related bottlenecks, which must be solved to develop very specifically targeting drugs for CTDSP1. Structure-based drug design has become a handy tool to solve many complex problems, such as handling the big data of combinatorial synthesis, improving the structure of multitarget drug molecules, dealing with the structural similarity of target proteins, and finding a perfect fit of drug molecules for target proteins in the drug discovery process [20,21,22,23,24,25,26,27,28,29,30,31,32,33,34,35]. Molecular docking is a standard modeling tool, and various docking techniques, such as allosteric docking, ensemble docking, and composite docking, have evolved to execute structurally complex issues [24,36,37,38,39,40,41,42,43,44,45,46,47,48,49,50,51,52,53,54,55,56,57,58,59,60,61,62]. Allosteric docking is a developing method to find drug molecules which are inhibiting or modulating through specific allosteric sites on the enzyme. Ensemble docking uses the generation of an ensemble of target protein, often gained from molecular dynamics simulation. Composite docking is a combined approach with several docking techniques. These docking techniques could provide a way of probing the druggability on the interface of protein–protein interactions [63], searching for the allosteric regulation site of target proteins [64,65], screening of target proteins with flexible structures [66,67], calculating the off-target binding of drug molecules [68,69], and overcoming the weaknesses associated with computer screening methods [70,71]. Therefore, we are going to explore the addressed issues and previous techniques followed by other researchers. We will focus on contemporary techniques of molecular docking, which will help make progress towards successful targeting of CTDSP1.

## 2. CTDSP1 as an Emerging Target

When selecting a new drug target, we must consider the activity of the target protein depending on various factors, such as expression level, subcellular localization, post-translational modification, and the protein–protein interaction network [72,73,74,75,76]. Previously, many experiments have acknowledged the various roles of CTDSP1, such as neuronal gene silencing, cell cycle regulation, and regulation of TGFβ and BMP signal transduction. The biological roles of CTDSP1 are summarized in Figure 1, Table 1, and another review paper [8]. Interestingly, recent studies have proclaimed new functional insights related to cancer, which is a new dimension of potential activity. Hence, here we are concentrating on how CTDSP1 is implicated in different cancer-related mechanisms and binding partners, as one of the new drug targets.

Cancer progression involves different stages of migration, invasion, and metastasis, and each step is carried out by different transcription factors. Among them, Twist-related protein (TWIST) is commonly identified in many cancers, such as breast, gastric, squamous cell carcinoma, cervical, ovarian, esophageal, and gastric cancers. TWIST is associated with different signaling cascades, like TGFβ and v-akt oncogene homolog (AKT)/phosphoinositide 3-kinase (PI3K), to promote cancer cell invasion and metastasis with the help of B lymphoma Mo-MLV insertion region 1 homolog (BMI1) and v-akt oncogene homolog 2 (AKT2) [77,78]. It has been observed that a mitogen-activated protein kinase (MAPK) phosphorylates the Twist-related protein 1 (TWIST1) on N-terminal Ser68, and stabilizes it to enhance cancer cell migration, invasion, and metastasis in breast cancer cells. Further, in contrast to the findings of previous studies, CTDSP1 has been shown to dephosphorylate the Ser68 residue of TWIST1 through the interaction of the N-terminus of TWIST1 and amino acid residues 43–63 of CTDSP1. Additionally, no other similar phosphatases were able to constitute this reaction with the same efficiency, and the inverse concentration of TWIST1 and CTDSP1 has been observed in several cancer cell lines. This suggests that CTDSP1 can potentially attenuate TWIST1 and antagonize cancer progression [16].

CTDSP1 has been known as a nuclear phosphatase, but in a recent study, researchers found that it is localized to the plasma membrane by lipid modifications [79]. This is also fashionably related to the tumor growth and N-terminal modification abilities of CTDSP1. AKT activation requires Ser473 and Thr308 phosphorylation, and consequently is involved in angiogenesis and tumor growth [80,81,82,83,84,85]. Many pieces of research have focused on the C-terminal activity of CTDSP1, but the understanding of N-terminal residues of CTDSP1 is evolving, with new activities identified in impeding tumor growth, invasion, and metastasis. Myristoylated AKT was found to be localized to the plasma membrane, and at the same time, CTDSP1 was also identified in the plasma membrane [86,87]. The reason for the membrane localization of CTDSP1 does not originate from a transmembrane domain, because CTDSP1 does not have the transmembrane domain [6]. N-terminal Cys44 and Cys45 of CTDSP1 (Figure 2) are the key residues involved in plasma membrane localization, which undergo the most dynamic posttranslational modification—palmitoylation [79]. CTDSP1 could be palmitoylated through the thioester bond between palmitate and N-terminal Cys, and the modification anchors CTDSP1 to the plasma membrane [79]. Mutational studies have demonstrated that only palmitoylated CTDSP1 performs selective dephosphorylation of AKT [79]. The palmitoylation-unable mutant of CTDSP1 failed in the removal of the phosphate group from the Ser473 of AKT [79]. The direct interaction of AKT and CTDSP1 could negatively modulate angiogenesis by tumor growth suppression through dephosphorylation of Ser473. These observations were illustrated by wound healing assay and in vitro experiments in HeLa Cells [79]. In summary, CTDSP1 can be a tumor suppressor through inhibiting cancer cell migration and invasion by the dephosphorylation of TWIST and AKT. The tumor suppressor properties of CTDSP2 and CTDSPL has also been reported in recent studies [88,89,90,91]. CTDSP2 and CTDSPL also inhibit tumor growth and angiogenesis through dephosphorylation of RB and tumor-suppressor protein promyelocytic leukemia (PML).

Given the above findings, CTDSP1 could be a novel target for cancer therapy. Other biological activities from earlier research have been based on the C-terminal domain and catalytic site of CTDSP1. Present experiments are showing that the N-terminal domain of CTDSP1 also has an inevitable role in CTDSP1 activity [79], which might be a unique feature compared to other CTDSPs. Thus, the regulation of CTDSP1 expression and activity may be related to the development of0 novel strategies for cancer treatment. However, there are still many challenges in targeting CTDSP1, such as solving the structural complexity among CTDSPs, finding the unique characteristics of CTDSP1, and looking for novel druggable materials to target CTDSP1 in cancer-related research and development.

## 3. Challenges and Opportunities in Targeting CTDSP1

There are seven human CTD phosphatases: CTDP1, CTDSP1, CTDSP2, CTDSPL, CTD small phosphatase like 2 (CTDSPL2), CTD nuclear envelope phosphatase 1 (CTDNEP1), and ubiquitin-like domain-containing CTD phosphatase 1 (UBLCP1). Based on the phylogenetic analysis of human CTD phosphatases presented in a previous article [96], CTDSP1, CTDSP2, CTDSPL, and CTDNEP1 are closely related, but CTDP1, UBLCP1, and CTDSPL2 are distantly related. Similar phenomena have been observed in a sequence alignment analysis presented in another review paper [95]. Currently, several structures of CTD phosphatases have been deposited into the protein data bank [6,7,17,97,98,99,100,101,102]. Among them, CTDSP1 (PDB ID: 2GHT, 2GHQ, 3PGL, 4YH1, 4YGY, 3L0B, 3L0C, 3L0Y, 2GHT, 1T9Z, 1TA0), CTDSP2 (PDB ID: 2Q5E), and CTDSPL (PDB ID: 2HHL) are structurally much conserved [5]. The structural similarity of CTDSP1, CTDSP2, and CTDSPL observed through the sequence and 3D structure alignment is presented in Figure 2. Additionally, the functionality of CTDSP2 and CTDSPL, such as their catalytic activity against RNAP II CTD peptide and their tumor-suppressing role, also overlaps with that of CTDSP1 [5,7,10,88]. However, CTDSP2 and CTDSPL have 10 or 12 residue insertions on the N-terminal end adjacent to the active sites, respectively, but CTDSP1 has no insertion on the N-terminal end, based on our structure pairwise comparison (Figure 2). Although they are operating on similar substrates, such as RNAP II CTD and R-SMADs with conserved signature active sites [5,7,10], they might be highly specific towards their binding partners due to N-terminal differences.

In earlier studies [17,18], researchers emphasized the structural complexity in targeting CTDSP as the significant bottleneck; hence, in this section, we are going to address the structural barriers and privileges of CTDSP1 as a successful biological target. The major challenge in targeting CTDSP1 is the conserved structures of CTDSP (Figure 2). CTDSP1, CTDSP2, and CTDSPL are approximately 40% conserved among their full sequences, and are highly conserved around the catalytic site [5,7,10]. Attacking the catalytic site of CTDSP1 may also impact other CTDSPs with identical domains [7,17]. The second complication is that CTDSP1 is implicated in different signaling events, such as neuronal gene silencing, negative regulation of cancer, and cell cycle regulation; hence, nonspecific targeting may impact many different biological activities of CTDSP1 [8]. However, CTDSP1 shows such different cellular localization traits that understanding the particular purpose of CTDSP1’s location in a cell is mandatory.

A thorough review of the literature demonstrated the history of selective inactivation of protein phosphatases with small molecules [103]. These studies have helped us learn about vital targetable sites, and will improve the precision in targeting CTDSP1. One successful instance occurred in impeding the CTDSP1 effect using rabeprazole, as a commercially available drug [17]. The co-crystallization experiment exhibited a precise targeting of CTDSP1 by using its unique features (Figure 3). The architecture of the CTDSP active site domain illustrates that the first aspartate residue (Asp96 in CTDSP1) acts as a nucleophile in the general acid/base mechanism. The consecutive one (Asp98 in CTDSP1) helps in the nucleophilic attack, along with metal ion Mg^2+^, but these residues are not involved in substrate binding. CTDSP1’s active site is surrounded by a unique hydrophobic pocket composed of Phe106, Val118, Leu155, Tyr158, and Arg178 residues (Figure 3B,D). It forms a binding groove with proline residues of a heptapeptide repeat of RNAP II CTD [6]. Rabeprazole blocks the binding groove and interrupts the dephosphorylation of CTD peptide [17]. The above result emphasizes that the hydrophobic pocket, shown in Figure 3, helps with the unique targeting of CTDSP1. Further, it can be inferred from previous studies [5,79] that the substrate selectivity of CTDSP1 will be initially decided by cellular localization and further binding groove residues. These characteristics will help with finding a novel targeting strategy to precisely regulate CTDSP1’s activity.

The field of drug designing has been rapidly advancing in recent decades, and many new state-of-the-art technologies and methods have been discovered [20,104,105,106,107,108,109,110,111,112,113,114]. Well rationalized structural information of receptor proteins and ligand chemicals is the essential requirement for drug designing. Based on this principle, the structure-based drug discovery (SBDD) process has evolved. In recent decades, the structure-based drug design technique has been considered the best tool to attack molecular targets with high accuracy. Highly ambiguous targets like enzymes need to be considered carefully and precisely. Doing so requires extensive structural information about the various ensembles, like pockets, folds, and hydrogen bonds. Focusing proteins in drug discovery direct the way to finding the best lead compound, with high efficacy at the lowest micromolar concentrations. Numerous software tools for protein-ligand dockings, such as Autodock, DOCK 4.0, GOLD, Flex-X, ICM, SLIDE, GLIDE, and DeepBindRG, were developed based on various scoring functions, including empirical, knowledge-based, and machine learning, and are summarized in Table 2 [50,115,116,117,118,119,120,121,122,123,124,125,126,127,128,129,130].

C-terminal domain small phosphatases have evolved with conserved structures and similar functions [5,96]. Proper structures for SBDD are available only for CTDSP1, CTDSP2, and CTDSPL. Finely determined X-ray structures (PDB ID: 2GHT, 2GHQ, 3PGL, 4YH1, 4YGY) are available for CTDSP1 in the PDB database. The essential requirement for structure-based drug design [131], which has also been considered in a few earlier studies [17,18], is the need to target precisely the successive regulation of CTDSP1. Previous studies can also help us to understand the essence of the combinatorial approach in the successful identification of novel small molecules for CTDSP1 as a drug target. The recent technology of allosteric docking has been tested successfully to find regulatory probe molecules for distinct molecules, like receptor tyrosine kinases (RTK) [132] and protein tyrosine phosphatases (PTP) [133]. Allosteric docking techniques to develop PTP inhibitors have been summarized in previous review papers [134,135]. We suggest the application of selective targeting techniques, such as allosteric targeting and ensemble docking, to overcome the obstacles in modulating the activity of CTDSP1.

Generally, people target the active site of a receptor protein to modify or inhibit the functionality of the receptor protein in drug discovery. However, it is very difficult to find a distinguished method for developing new molecules for biological or metabolic target proteins with structural similarities. Traditional inhibition techniques will harm the system by targeting active sites [105]. Other functionally imperative sites can be identified through allosteric targeting, apart from the regular active site of a receptor protein. The identified allosteric sites can help restrain the activity of a protein. Allosteric targeting can be achieved in numerous ways, such as targeting protein–protein interactions, focusing on multivalent inhibitors, and targeting disulfide bonds [136,137,138,139,140]. In the process of allosteric targeting, pharmacophore modeling of a ligand could apply for targeting both a directed specific site and other distinctive sites of a protein. Therefore, controlling protein activity by selecting allosteric sites of a receptor protein could be one of the targeting methods for CTDSP1.

In recent drug discovery history, with the advancement of technology, robust tools have been evolving, and some researchers are aiming to control activities at the metabolic level by targeting checkpoint signaling molecules in cancer [141,142]. The allosteric docking technique gained importance, since these vital targets were operated effectively. Allosteric docking involves a few crucial steps like allosteric binding site identification, analysis, and optimization [104,105,143]. The allosteric binding sites are identified through the literature and database searches, cavity finding, blind docking, coevolution, and so forth. Optimized allosteric binding sites are obtained from analysis and optimization, with the help of molecular dynamics simulation through perturbation response scanning, trajectory analysis, course-grained residue network analysis, and calculating Gibbs free energy [144]. Allosteric docking involves similar docking tools and algorithms to the other docking techniques summarized in Table 2. A careful and precise selection of allosteric binding sites based on understanding biochemical and biological characteristics of target proteins, and choosing proper docking tools, could increase the success rate of targeting [145,146]. A study on protein tyrosine phosphatase 1B (PTP1B), which is a crucial member in various biological activities and has the conserved catalytic site among PTP, showed that an allosteric inhibitor causes conformational changes in the protein outside the active site of PTP1B, and abolishes the enzyme activity of PTP1B extrinsically [147].

Similarly, CTDSP1 also has a conserved structure and similar activity to other CTDSPs. We observed that CTDSP1 also possesses characteristics from co-crystallized ligand experiments and a unique hydrophobic site adjacent to the active site, shown in Figure 3, which helps the inhibitor to bind to CTDSP1 without affecting catalytic residues [17,18]. Interestingly, CTDNEP1, as one of the conserved paralogs, was not affected by the ligand rabeprazole as much as CTDSP1 [17]. This report suggests that the hydrophobic pocket of CTDSP1 is highly specific for CTDSP1’s substrate recognition, and has importance in dephosphorylation activity against natural substrates. This site could be a potential binding target for allosteric drugs, which is satisfied with the fundamental requirement for non-conventional targeting. Furthermore, finding an allosteric targeting site for CTDSP1 is so recent that there might be many chances to identify novel binding sites to target CTDSP1 allosterically.

Observations from recent studies [16,79] suggest that N-terminal palmitoylated CTDSP1 is required for TWIST1 protein dephosphorylation. This whole mechanism is constituted through palmitoylation of the Cys44 and Cys45 residues on an N-terminal domain of CTDSP1 (Figure 1 and Figure 2). The mechanism is also specific to CTDSP1, since no other paralogs were evidenced in the experiment. This observation could be one of the other potential targetable allosteric sites. Although the binding possibility of allosteric drug molecules of CTDSP1 to other CTDSP has not been totally excluded, we may develop a way of targeting the N-terminal region of CTDSP1 based on the difference of N-terminal sequences among CTDSPs. In this case, the ligand might be bound away from the catalytic site of CTDSP1, and the targeting chemicals might not influence all other CTDSP members. Thus, allosteric docking is one of the appropriate tools for selective targeting of CTDSP1 using several predicted sites or regions to be found through a careful investigation.

Ensemble docking is an emerging method that works based on the structure-based drug design technique. It is a high-throughput and robust drug discovery technique, where multiple structures can be validated simultaneously and accurately. This docking method uses all discrete ensembles of a protein structure (EPS) [49,148]. In the ensemble docking process, generating various conformations of a receptor protein is the first step, and then, degrees of freedom will be calculated based on ligand binding. GOLD docking software is extensively used for implementing this technique. Initially, the structural library of a target protein with its conformational structures is prepared, and then, it is compared with a highly converged stable reference structure. Root mean square deviation (RMSD) differences of molecular dynamics trajectories based on the comparison with the reference structure are often calculated in vital regions of a target protein [149]. The structures are docked with the desired ligand, and conformational binding poses are predicted for complexes. Based on the degree the experimental structure is conserved, predicted poses would be ranked and continued for post-processing. This technique has been successfully applied with the p38 MAP kinase system, and outperformed conventional methods [150]. This method could be the best for targeting CTDSP1 due to the structural similarities among CTDSPs, the existence of multiple structures that can be compared simultaneously, and the availability of experimental data of co-crystallized ligands for accurate pose prediction and analysis.

Comparison of the chemical and natural ligand structure’s discretion against the apo-protein demonstrates the stability of the receptor protein and differences in conformational changes, which are useful for validating other paralog structures [46,151]. The RMSD values of the backbone and side-chain atoms denote the fundamental differences among protein structures, and extend insights into developing selective inhibitors [152]. Ensemble docking is one of the most promising techniques for modulating CTDSP1 activity, since CTDSP1, CTDSP2, and CTDSPL have the well-built X-ray structures (Figure 2B) required for ensemble prediction and analysis. Moreover, CTDSP1 has multiple 3D structures (Figure 3E) incorporated with natural and chemical ligands, for use as a reference structure. Therefore, ensemble docking can elicit the information of unique sites on CTDSP1 compared to other CTDSPs, which is vital for drug discovery targeting CTDSP1.

We present the comparison of conventional docking, allosteric docking, and ensemble docking of targeting CTDSP1 in Figure 4. Conventional docking target the active site of CTDSP1, disturbing the diverse biological roles of it and the catalytic activities of CTDSP2 and CTDSPL. Thus, conventional docking is not a desirable method of SBDD. Instead of the conventional approach, we suggest allosteric docking and ensemble docking for targeting CTDSP1 as better methods of SBDD. Allosteric docking targets a binding site other than the active site, which can, therefore, produce much more specific drug molecules for CTDSP1 without interrupting the activities of other CTDSPs. Ensemble docking uses the generated conformational library of CTDSP1 and produces the most vital regions of it. The produced essential regions could be tested through biochemical screening, and may contribute optimization when targeting CTDSP1. However, we agree that there could be yet unexplored, better docking methods than allosteric docking and ensemble docking in the targeting of CTDSP1.

## 4. Conclusions

CTDSP1’s protein activity is vital for various biological activities. To date, the C-terminal catalytic domain of CTDSP1 has been studied, and limited activities of CTDSP1 identified. Interestingly, recent studies have extended the knowledge we have of CTDSP1’s functionality by focusing on the N-terminal region of CTDSP1. This new functional insight, as a novel protein regulator, has increased the significance of CTDSP1. The highly similar structures of CTDSPs have become barriers to targeting CTDSP1. Therefore, we focused on finding contemporary and robust methods and tools to confront this structural problem. The previous studies offered information about a critical functional site of a hydrophobic pocket adjacent to the active site of CTDSP1, which is essential for accommodating natural substrates. The N-terminal region of CTDSP1 is required for palmitoylation, to anchor proteins in the plasma membrane, and was identified as a potential targetable allosteric site of CTDSP1. A novel allosteric site may also be found through the investigation on protein–protein interaction analysis of CTDSP1. Targeting the hydrophobic pocket, N-terminal region, and a novel site of CTDSP1 could provide a new chance to develop cancer-related treatments. Allosteric docking has been identified as a suitable technique for this, as it can precisely target desired sites on receptor proteins. Therefore, allosteric docking could be an approach to target the hydrophobic pocket, N-terminal region, and novel site of CTDSP1. Ensemble docking produces numerous structural confirmations of receptors through molecular dynamics. The generated ensembles help to exclude false positives and discover highly specific inhibitors. CTDSP1 has close structural analogues, like CTDSP2 and CTDSPL. Rich data are available on CTDSP1’s structure, which will be very useful for the design of potential novel drugs through ensemble docking with CTDSP1. Thus, the application of ensemble docking for CTDSP1 could potentially outperform conventional methods. Finally, selectivity could be achieved by associating allosteric docking and ensemble docking as robust and high-throughput methods for targeting CTDSP1. In order to develop particular drug molecules for CTDSP1, more detailed investigations searching for biochemical characteristics among CTDSPs, revealing the protein–protein interaction network of CTDSP1, and finding a novel allosteric site of CTDSP1 are required. We hope that more researchers will continue to disclose more precise ways of targeting CTDSP1, with the help of structure-based drug design.

## Figures and Tables

**Figure 1 life-10-00057-f001:**
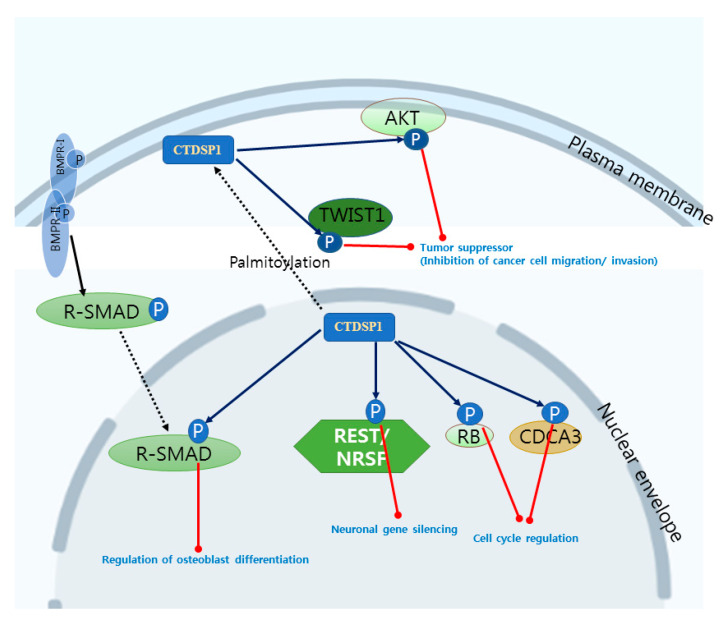
Scheme illustrating the biological substrates and roles of CTDSP1. The dashed arrows represent subcellular translocation of proteins, and the solid arrows represent phosphorylation (black) and dephosphorylation (blue) of proteins. The red lines indicate the biological results of dephosphorylation by CTDSP1.

**Figure 2 life-10-00057-f002:**
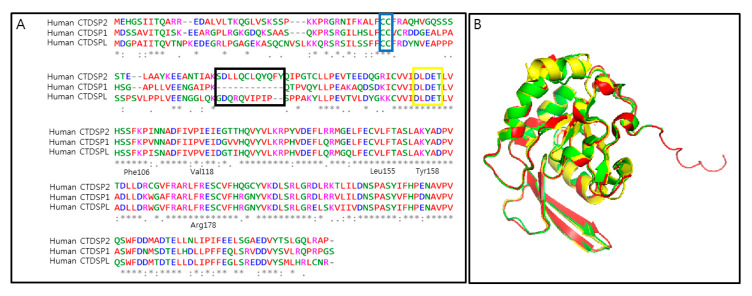
Structural similarity of CTD small phosphatases. (**A**) The sequence alignment of human CTDSP1 1-261 (NCBI accession number: NP_067021.1) with human CTDSP2 1-271 (NCBI accession number: NP_005721.3) and human CTDSPL 1-276 (NCBI accession number: NP_001008392.1), showing the active site (yellow square), palmitoylation residues (blue square), N-terminal insertion residues (black square), and representative residues consisting of the hydrophobic pocket of human CTDSP1 written in black below the alignment. (**B**) 3D structure alignment of human CTDSP1 (green, PDB ID: 3PGL), human CTDSP2 (yellow, PDB ID: 2Q5E), and human CTDSPL (red, PDB ID: 2HHL).

**Figure 3 life-10-00057-f003:**
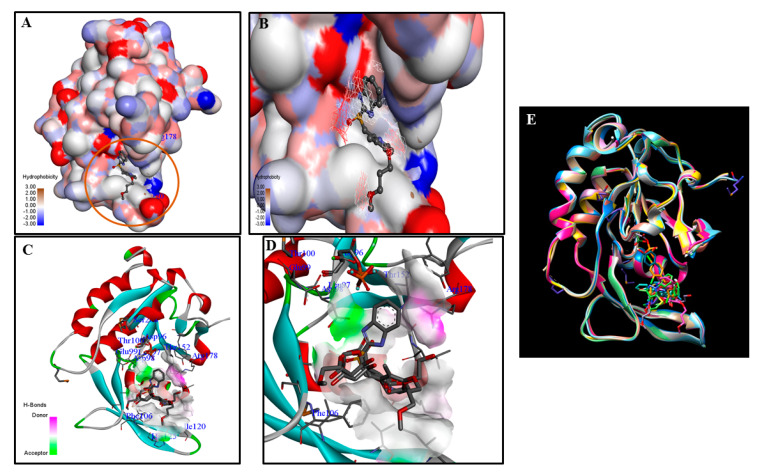
Structural representation of human CTDSP1. (**A**) The hydrophobic pocket of human CTDSP1 along with the co-crystallized ligand rabeprazole, marked with an orange circle, adapted from the PDB database (PDB ID: 3PGL). (**B**) Close-up view of the hydrophobic pocket in A. (**C**) The active site and hydrophobic pocket of human CTDSP1, along with rabeprazole and CTD phosphopeptide, showing the related amino acid residues. (**D**) Close-up view of the active site and hydrophobic pocket in C. (**E**) Clustered structures of human CTDSP1 along with natural and chemical ligands adapted from the PDB databank (PDB ID: 2GHT, 2GHQ, 3PGL, 4YH1, 4YGY).

**Figure 4 life-10-00057-f004:**
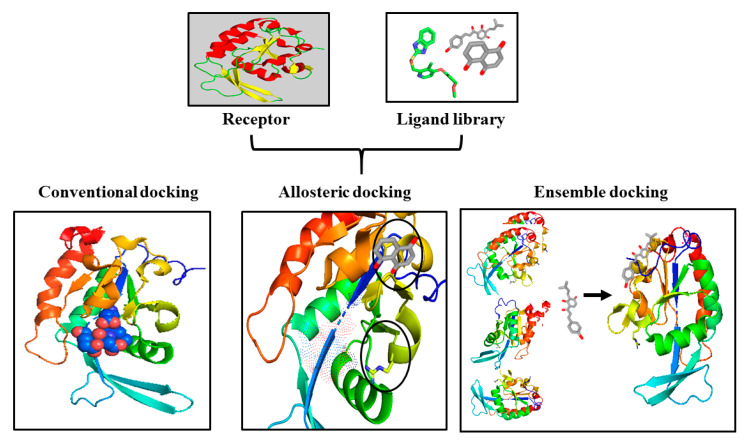
Comparison of some molecular docking methods. The receptor protein shows the structure of human CTDSP1, and the ligand library shows several candidate compounds [17,18,153] for targeting CTDSP1. The left square represents conventional docking, which targets the active site of human CTDSP1. The middle square shows allosteric docking, which finds functionally significant regions as allosteric sites, shown with black circles, of human CTDSP1. The right square presents the ensemble generation of human CTDSP1, which is the primary technique used in ensemble docking.

**Table 1 life-10-00057-t001:** Biochemical characteristics and biological roles of CTD small phosphatases, Modified from another review paper [92].

CTD Small Phosphatase	Aliases	Substrates	PDB IDs	Biological Roles [References]
CTDSP1	SCP1 NIF3	RNAP II CTD REST R-SMADs CDCA3, RB TWIST1, AKT PML, c-Myc	4YGY, 4YH1, 3PGL, 3L0B, 3L0C, 3L0Y, 2GHQ, 2GHT, 1T9Z, 1TA0, 3PGL	Transcription factor recruitment [5,93] Neuronal gene silencing [9,10] Osteoblast differentiation [12] Cell cycle regulation [13,14] Tumor suppressor [88,89,90,91]
CTDSP2	SCP2 OS4 NIF2	RNAP II CTD R-SMADs PML	2Q5E	Transcription factor recruitment [5,94] Promoter clearance [94] Ras activation [95]
CTDSPL	SCP3 HYA22 NIF1	RNAP II CTD R-SMADs PML	2HHL	Transcription factor recruitment [5] Tumor suppressor [90]

**Table 2 life-10-00057-t002:** Comparison of some frequently-used molecular docking tools.

Docking Tool	Algorithm	Scoring Function	Representative References
Autodock	Genetic algorithm, Simulated annealing, Lamarckian algorithm	Force field, Empirical	[55]
DOCK 4.0	Incremental search, Shape fitting	Force field	[54]
GOLD	Genetic algorithm	Force field	[57]
Flex-X	Multiple copy, Simultaneous search, Incremental search	Empirical	[58]
ICM	Monte Carlo sampling	Empirical	[59]
SLIDE	Incremental construction	Force field, Empirical	[60]
GLIDE	Monte Carlo sampling	Empirical	[61,62]
GOLD/ASP	Genetic algorithm	Knowledge-based	[127,128]
DeepBindRG	Deep learning algorithm	Machine learning	[129]

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
