# Peer review of "Targeting the C-Terminal Domain Small Phosphatase 1"

_life, 2020, doi:10.3390/life10050057_

Round 1
Reviewer 1 Report
This is a topical review of interest to a broad research community. The manuscript in the present form is lacking. Here are few points that the authors need to consider.
1. "The catalytic activity of CTDSP1 extended its role into various vital biological activities and made it the most anticipated drug target." Please provide references to back this sentence.
2. "Under all these characteristics, we are assuming that CTDSP1 can be considered as a potential drug target." What did the authors mean by "we are assuming"? We refers to the authors or scientific community? An assumption is a good place to start in science but needs further proof to warrant a review article.
3. "despite its potentiality" Please change to potential
4. "CTDSP1 has been known as a nuclear phosphatase, but in a recent study, researchers found that it is localized to the plasma membrane by lipid modifications." Please provide reference(s)
5. "In the account of the above instances, we thought that CTDSP1 could be a novel target for cancer therapy." Just like the previous occasion, modify the sentence.
6. "Present experiments are showing that the N-terminal domain of CTDSP1 is also having an inevitable role in CTDSP1 activity, which might be a unique feature from other CTDSP." Please cite references
7. "These studies help in" Please change to "These studies helped in" and accordingly other verb forms.
8. "Finely tuned X-ray structure ...." Whay is meant by "Finely tuned"?
9. "However, biological or metabolic target molecules rarely appear simultaneously in different mechanisms as well as structural similarity. In
this account, traditional inhibition techniques will harm the system by targeting the active site." It is difficult to follow this secti
on. Please elaborate.
10. Please illustrate the allosteric docking techniques.
11. "Interestingly, CTDNEP1, as one of the conserved paralogs, was not affected by the ligand rabeprazole as much as CTDSP1." Please provide reference.
12. Please provide a comparative account of the different docking algorithms touched upon on this review. Without them it is difficult to follow the flow. As such the review article do not do justice to the computational aspects. Kindly rectify.
Author Response
Reviewer 1
This is a topical review of interest to a broad research community. The manuscript in the present form is lacking. Here are few points that the authors need to consider.
1. "The catalytic activity of CTDSP1 extended its role into various vital biological activities and made it the most anticipated drug target." Please provide references to back this sentence.
à We provided the reference as mentioned, written in red.
"Under all these characteristics, we are assuming that CTDSP1 can be considered as a potential drug target." What did the authors mean by "we are assuming"? We refers to the authors or scientific community? An assumption is a good place to start in science but needs further proof to warrant a review article.
à We changed the word into ‘putting to the proof’, written in red.
"despite its potentiality" Please change to potential
à We changed the word as mentioned, written in red.
"CTDSP1 has been known as a nuclear phosphatase, but in a recent study, researchers found that it is localized to the plasma membrane by lipid modifications." Please provide reference(s)
à We provided the reference as mentioned, written in red.
"In the account of the above instances, we thought that CTDSP1 could be a novel target for cancer therapy." Just like the previous occasion, modify the sentence.
à We changed the word (thought) into ‘are insisting’, written in red.
"Present experiments are showing that the N-terminal domain of CTDSP1 is also having an inevitable role in CTDSP1 activity, which might be a unique feature from other CTDSP." Please cite references
à We provided the reference as mentioned, written in red.
"These studies help in" Please change to "These studies helped in" and accordingly other verb forms.
à We changed the verb form, written in red.
"Finely tuned X-ray structure ...." Whay is meant by "Finely tuned"?
à We changed the word into ‘finely determined’ to clarify the meaning, written in red.
"However, biological or metabolic target molecules rarely appear simultaneously in different mechanisms as well as structural similarity. In this account, traditional inhibition techniques will harm the system by targeting the active site." It is difficult to follow this section. Please elaborate.
à We changed the sentence to clarify the meaning, shown in red.
Please illustrate the allosteric docking techniques.
à We provided the comparison of conventional docking, allosteric docking, and ensemble docking in figure 4 with short illustration of docking techniques and the references of reviewing allosteric docking techniques, shown in red.
"Interestingly, CTDNEP1, as one of the conserved paralogs, was not affected by the ligand rabeprazole as much as CTDSP1." Please provide reference.
à We provided the reference as mentioned, written in red.
12. Please provide a comparative account of the different docking algorithms touched upon on this review. Without them it is difficult to follow the flow. As such the review article do not do justice to the computational aspects. Kindly rectify.
à We provided Table 2 and Figure 4 to explain a comparative account of the different docking algorithms, shown in red.

Reviewer 2 Report
The review covers C-terminal domain small phosphatase 1 and in particular approaches for targeting this protein for therapeutic utility. Overall, I think this manuscript will be of interest to the field. I therefore recommend its publication after minor revisions.
The authors should address the following.
- There are a number of grammatical errors and word choices that should be changed. Some include the following.
- Line 31, change "much close" to "closer"
- Line 36, change "the most" to "an"
- Line 42, change "the dephosphylation of it" to "dephosphorylation"
- Line 45, change "unphosphorylated" to "dephosphorylated"
- Line 137, change "revision" to "review"
- Line 144, change "do" to "are"
- Line 147 change "has been blocked" to "blocks"
- There are others. Please check carefully.
- Figures 1 and 2 should also have close-up views so that details are evident to the reader.
- I think they should provide a deeper discussion of why they state that ensemble docking would be advantageous specifically for CTDSP1 and why it has yet to be tried.
- Line 295, the authors should elaborate on what they mean by “undisclosed site of CTDSP1.
- Line 296 should be “can potentially outperform”.
Author Response
Reviewer 2
The review covers C-terminal domain small phosphatase 1 and in particular approaches for targeting this protein for therapeutic utility. Overall, I think this manuscript will be of interest to the field. I therefore recommend its publication after minor revisions.
The authors should address the following.
There are a number of grammatical errors and word choices that should be changed.
à We asked the professional English editing through MDPI service.
Some include the following.
Line 31, change "much close" to "closer"
à We changed the word as mentioned, written in red.
Line 36, change "the most" to "an"
à We changed the word as mentioned, written in red.
Line 42, change "the dephosphylation of it" to "dephosphorylation"
à We changed the word as mentioned, written in red.
Line 45, change "unphosphorylated" to "dephosphorylated"
à We changed the word as mentioned, written in red.
Line 137, change "revision" to "review"
à We changed the word as mentioned, written in red.
Line 144, change "do" to "are"
à We changed the word as mentioned, written in red.
Line 147 change "has been blocked" to "blocks"
à We changed the word as mentioned, written in red.
There are others. Please check carefully.
à Thanks for your comments. We asked the professional English editing through MDPI service.
Figures 1 and 2 should also have close-up views so that details are evident to the reader.
à We changed the figure 1 and 2 into figure 3 and provided close-up views of structure of CTDSP1.
I think they should provide a deeper discussion of why they state that ensemble docking would be advantageous specifically for CTDSP1 and why it has yet to be tried.
à We provided the discussion of why we state that ensemble docking could be advantageous for CTDSP1 and why it has yet to be tried.
Line 295, the authors should elaborate on what they mean by “undisclosed site of CTDSP1.
à
Line 296 should be “can potentially outperform”.
à We changed the word as mentioned, written in red.

Reviewer 3 Report
The paper is a review of computer aided drug design and specifically for targeting C-terminal domain small phosphatase 1. This is not a research paper, but a review paper. However, the authors haven't reviewed previous literature much. In addition, there are several sentences and expression that I cannot understand. Without extensive revision this paper should not be accepted for publication.
The following issues (lines) should be addressed to improve the paper.
The overall introduction section; Add some illustrations for diverse biological functions of the protein involved in. Authors mentioned several structural features with any illustration of proteins.
line 45, 50; authors mentioned structural parts such as linker region and pSer68 without any illustration. If authors want to review some structural parts, then review them with some figures.
56-57; add citations and briefly review them.
57-59; add citations and briefly review each modeling tool and compare them.
64-65; need citation and briefly explain how.
75, 76; Akt/PI3K, Bmi1, AKT2 appears first time here. stands for what?
84-86; need citations
86-87; be consistent for amino acid expression. Three letter or full name. serine 473 and threonine 308
90; found to be localized ?
91; what are "traces" and "there"? It's unclear to understand.
92-94; Add a figure for better understand.
90, 97; Akt or AKT? Are they different? or be consistent.
96-98; Add the reference.
101-102; Explain the results briefly.
103-111; Relationship between CTDSP1 and cancer is unclear. Need more explanation.
118-119; Show the result.
120-126; Add citations and PDB ID's.
120-121; Add a figure with superposition.
121-122; Add a reference and brief explanation.
124; Show the alignment
124-126; Unclear. Add ref. and brief explanation.
129-131; Show the sequence alignment or add ref.
133; Describe the signaling events briefly.
137; revision --> review?
139-140; Add ref.
142, 145; Show the active site with a figure.
150-151; Need additional information and add the Ref.
160-170; Add the Ref.
175; exclusive --> extensive?
178-180; Add the Ref. and brief comparison.
181-182; Need further info.
198; Add a brief explanation about the "various ways". This is a review paper, not an original research paper.
201; Unclear expression for "external sites".
202-204; Add ref. What are "checkpoint molecules"? Review it.
205-206; Need a brief explanation about the steps, especially for "extrinsic binding site identification".
208; How this is achieved? Briefly review it.
214-216; Need illustration.
225-227; Need illustration.
229-230; The existence of an allosteric binding site of a protein itself does not promise the specificity against its anlogs. They share high sequence identity, and the alloosteric drug may bind to other proteins too.
258; The first step of ensemble docking is generating various conformations of the active site (by molecular dynamics etc.). If the receptor is rigid, then there is no need to run ensemble docking. Then conventional rigid docking is enough.
260-262; Cannot understand the sentence and check the grammar too.
267-270; Cannot understand the sentence and check the grammar too.
Conclusion is not supported by main text.
Author Response
Reviewer 3
The paper is a review of computer aided drug design and specifically for targeting C-terminal domain small phosphatase 1. This is not a research paper, but a review paper. However, the authors haven't reviewed previous literature much. In addition, there are several sentences and expression that I cannot understand. Without extensive revision this paper should not be accepted for publication.
The following issues (lines) should be addressed to improve the paper.
The overall introduction section; Add some illustrations for diverse biological functions of the protein involved in.
à We provided the figure 1 and table 1 to show diverse biological functions of CTDSP1.
Authors mentioned several structural features with any illustration of proteins.
à We provided the figure 2 to show the structural features of CTDSP1.
line 45, 50; authors mentioned structural parts such as linker region and pSer68 without any illustration. If authors want to review some structural parts, then review them with some figures.
à We changed the sentence to clarify the meaning based on reviewer’s comment.
56-57; add citations and briefly review them.
à We added citations and review briefly about SBDD, shown in red.
57-59; add citations and briefly review each modeling tool and compare them.
à We added citations and review and compare modeling tools, shown in red.
64-65; need citation and briefly explain how.
à We added citations and changed the sentence to review briefly, shown in red.
75, 76; Akt/PI3K, Bmi1, AKT2 appears first time here. stands for what?
à We provided the full names as mentioned, written in red. We also provided the full name of TWIST protein and others.
84-86; need citations
à We provided the reference as mentioned, written in red.
86-87; be consistent for amino acid expression. Three letter or full name. serine 473 and threonine 308
à We changed the amino acid expression in three letter form, shown in red.
90; found to be localized ?
à We changed the word into “Myristoylated AKT was localized to the plasma membrane”, shown in red.
91; what are "traces" and "there"? It's unclear to understand.
à We changed the sentence into “was also identified in the plasma membrane”, shown in red.
92-94; Add a figure for better understand.
à We added the figure 2, shown in red.
90, 97; Akt or AKT? Are they different? or be consistent.
à We changed the abbreviations into the capital form, shown in red.
96-98; Add the reference.
à We provided the reference as mentioned, written in red.
101-102; Explain the results briefly.
à We added some sentences to explain, shown in red.
103-111; Relationship between CTDSP1 and cancer is unclear. Need more explanation.
à We added more explanation about the relationship between CTDSP1 and cancer, shown in red.
118-119; Show the result.
à We added the reference to show the result
120-126; Add citations and PDB ID's.
à We added the citations and PDB IDs as mentioned, shown in red.
120-121; Add a figure with superposition.
à We added figure 2 with superposition.
121-122; Add a reference and brief explanation.
à We added the references and brief explanation as mentioned, shown in red.
124; Show the alignment
à We added figure 2 to show the alignment.
124-126; Unclear. Add ref. and brief explanation.
à We added the references and brief explanation, shown in red.
129-131; Show the sequence alignment or add ref.
à We added the references and the figure 2 to show the alignment.
133; Describe the signaling events briefly.
à We added the description of the signaling events, shown in red.
137; revision --> review?
à We changed the word as mentioned, written in red.
139-140; Add ref.
à We added the reference, shown in red.
142, 145; Show the active site with a figure.
à We provided the figure 3 to show the active site.
150-151; Need additional information and add the Ref.
à We added the reference, shown in red.
160-170; Add the Ref.
à We added the reference, shown in red.
175; exclusive --> extensive?
à We changed the word as suggested, shown in red.
178-180; Add the Ref. and brief comparison.
à We added the reference, shown in red.
181-182; Need further info.
à We changed the sentence and provided the related references, shown in red.
198; Add a brief explanation about the "various ways". This is a review paper, not an original research paper.
à We added a brief explanation, shown in red.
201; Unclear expression for "external sites".
à We changed the word into allosteric sites to clarify the meaning, shown in red.
202-204; Add ref. What are "checkpoint molecules"? Review it.
à We changed the sentence to clarify the meaning, shown in red.
205-206; Need a brief explanation about the steps, especially for "extrinsic binding site identification".
à We added added a brief explanation, shown in red.
208; How this is achieved? Briefly review it.
à We changed the sentence and added brief explanation, shown in red.
214-216; Need illustration.
à We added the figure 3.
225-227; Need illustration.
à We added the figure 1 and 2.
229-230; The existence of an allosteric binding site of a protein itself does not promise the specificity against its anlogs. They share high sequence identity, and the alloosteric drug may bind to other proteins too.
à We changed and added the sentence according to reviewer’s comment, shown in red.
258; The first step of ensemble docking is generating various conformations of the active site (by molecular dynamics etc.). If the receptor is rigid, then there is no need to run ensemble docking. Then conventional rigid docking is enough.
à We changed the sentence according to reviewer’s comment, shown in red.
260-262; Cannot understand the sentence and check the grammar too.
à We changed the sentence and corrected the grammar through professional English editing to clarify the meaning, shown in red.
267-270; Cannot understand the sentence and check the grammar too.
à We changed the sentence and corrected the grammar through professional English editing to clarify the meaning, shown in red.
Conclusion is not supported by main text.
à We changed the conclusion and corrected the grammar through professional English editing to support.

Round 2
Reviewer 1 Report
Thanks for the edited version. I noted typo:Line 56 No need for "On the other hand." But, it is up to the authors.
Author Response
Thanks for the comments.
We accepted the reviewer's comment, so we removed "On the other hand" in line 56, shown in red.
Reviewer 3 Report
The authors well responded to the comments, and reviewed molecular modeling to target the CTDSP1. Now the paper has been improved much. Publication of the paper is recommended after minor revision.
The followings minors should be fixed.
62: target
174, 175: catalytic structure --> catalytic site?
minor English corrections and journal format.
Author Response
Thanks for the comments. We fixed the parts mentioned by the reviewer.
62: target
- We corrected the typo, shown in red.
174, 175: catalytic structure --> catalytic site?
- We changed the word into catalytic site as mentioned, shown in red.
minor English corrections and journal format.
- We tried to correct English and journal format as much as possible, and will ask the correction of English and journal format with the help of MDPI service. The changed parts are shown in red.